# Complexity and accessibility of random landscapes

**Sakshi Pahujani⋆ and Joachim Krug†**

Institute for Biological Physics, University of Cologne, Köln, Germany

⋆ spahuja1@uni-koeln.de , † jkrug@uni-koeln.de

Part of the *2024-07: Theory of Large Deviations and Applications collection*
*Session 123 of the Les Houches School, July 2024*
*published in the Les Houches Summer School Lecture Notes series*

## Abstract

These notes introduce probabilistic landscape models defined on high-dimensional discrete sequence spaces. The models are motivated primarily by fitness landscapes in evolutionary biology, but links to statistical physics and computer science are mentioned where appropriate. Elementary and advanced results on the structure of landscapes are described with a focus on features that are relevant to evolutionary searches, such as the number of local maxima and the existence of fitness-monotonic paths. The recent discovery of submodularity as a biologically meaningful property of fitness landscapes and its consequences for their accessibility is discussed in detail.

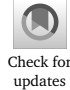

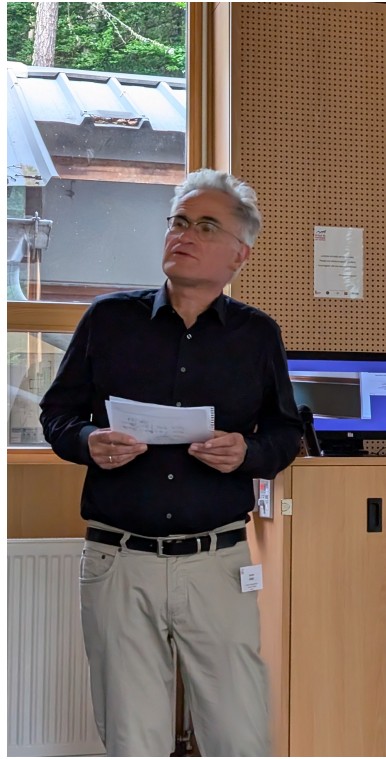



SciPost Phys. Lect. Notes 108 (2025)

# 1 Introduction

Navigation on high dimensional landscapes is a theme that presents itself in several fields. This is simply because real world systems are complex many-body systems. A study of the dynamics of such systems therefore involves functions of many variables, be it a free energy function in disordered systems like spin glasses, cost functions in optimisation problems defined on neural networks or fitness functions quantifying the reproductive success of organisms [1, 2]. In such settings, the stationary points of these functions or landscapes are particularly of interest as they signify a potentially stable state of the system. Depending on how the landscape is constructed, which in turn depends on the specific context, the landscapes may feature multiple stationary points. At first glance, one might infer that navigating to a globally stable point would be difficult owing to a tendency to get trapped in metastable states. Yet, as we will later explore in the text, this is not always true. Other properties that have piqued the interest of many are the dependence of accessibility of a stationary point on the initial state (i.e. the basin of attraction) and the distribution of the stationary points of a landscape on its domain.

    Naturally over the years many advances have been made towards characterising the properties of such stationary points. In this note we deliver an account of some of these properties in the context of biological evolution towards maxima of the fitness function, hereafter re-

ferred to as peaks, driven by natural selection. As we will shortly see, such a fitness function is defined on a combinatorially large space. Two approaches to define a fitness function on this space are discussed. The first approach assigns fitness values drawn randomly from a distribution to elements of the domain. In this framework, the statistics of the number of peaks are studied and their accessibility characterised. Then we do the same for structured fitness landscapes where fitness values are correlated to some extent. An important motivation for the study of structured fitness landscapes is the observation that empirical biological fitness landscapes display widely different degrees of ruggedness, and therefore cannot generally be well represented by an uncorrelated random model [3–5]. We will return to this point in Section 3.

The models and methods used to arrive at the results presented here are often borrowed from, closely associated with or may also potentially inspire techniques to study high dimensional landscapes in other contexts. For example, the House of Cards model discussed in section 4 is analogous to the exactly solvable Random Energy Model [6] of spin glasses. Another such connection is stated in the form of a map from a structured fitness landscape – Fisher's geometric model [7,8] – to the antiferromagnetic Hopfield model [9], see Sect. 6.3 for details. So while the mathematical structures employed to study fitness landscapes are enticing in their own right, these associations affirm their relevance beyond the confines of this text.

Finally, a small disclaimer– as in the short lectures on which this text is based, expressions for certain results in the following sections are taken from the literature without providing proofs. While readers are encouraged to approach these results as gateways to deeper inquiry, appreciating their role within the broader context of this note will suffice for the most part.

## 2 Definitions

We begin by establishing the necessary definitions:

- **Genotype.** A genotype is the hereditary information passed on from a parent to its offspring. Such information is carried by the nucleic acid polymers DNA and/or RNA. Here, we represent a genotype by a sequence $\sigma$ of length $L$,

$$\sigma = (\sigma_1, \sigma_2, \sigma_3, ..., \sigma_L), \tag{1}$$

  where the element $\sigma_i$ at a *site $i$* of the sequence is taken from an alphabet of size $a$ represented in this note by $0, 1, 2, ..., a-1$. Each element of the alphabet set is called an **allele**. The sequence space is therefore the $L$-fold Cartesian product of the set of alleles,

$$\sigma \in \{0, ..., a-1\}^L. \tag{2}$$

- **Hamming distance.** For a given pair of sequences $\sigma$ and $\tau$, the Hamming distance $d(\sigma, \tau)$ is defined as the number of differing sites between the two,

$$d(\sigma, \tau) = \sum_{i=1}^{L} (1 - \delta_{\sigma_i, \tau_i}). \tag{3}$$

- **Hamming graph.** A set of sequences of length $L$, $\{0, ..., a-1\}^L$ equipped with the Hamming distance measure is a Hamming graph, $\mathbb{H}_a^L$. The binary Hamming graph, $\mathbb{H}_2^L$ is a hypercube in $L$ dimensions. Figure 1 depicts such hypercubes for $L = 1, 2$ and $3$.

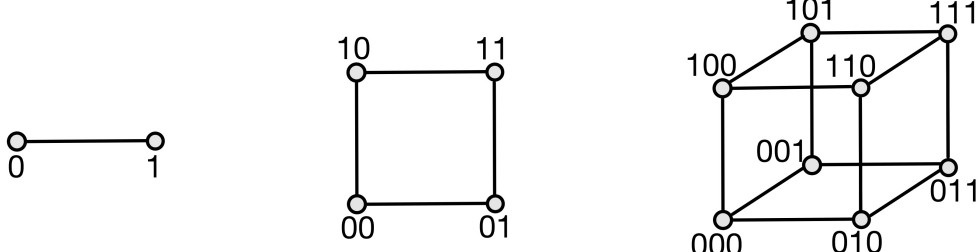

Figure 1: **Binary Hamming graphs.** From left to right, the figure shows hypercubes of dimension $L = 1, 2$ and $3$.

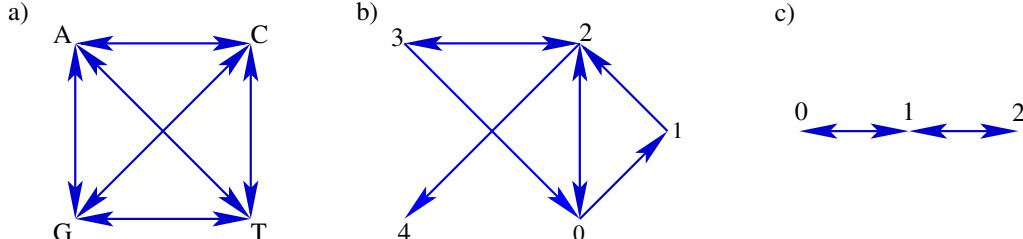

Figure 2: **Examples of allele graphs.** (a) The allele graph of nucleotides (constituents of DNA). (b) An incomplete allele graph over five alleles. This is an example of an allele graph for which the bound $\beta^*$ on the accessibility threshold $\beta_c$ discussed in Sect. 4.2.2 is not tight, i.e., $\beta_c > \beta^*$. (c) An allele graph over three alleles (linear path graph). For this graph $\beta^* > 1$, which implies that there are no (direct or indirect) accessible paths for any fitness difference between initial and endpoint genotype. See [10] for details.

- **Mutations.** In general mutations are any form of changes in the genotype. In this note, we will consider only *point mutations* which simply change the allele at one site of a genotype $\sigma$:

$$\sigma_i \rightarrow \sigma_i'. \tag{4}$$

- **Allele graph.** As such, Hamming graphs are constructed with an underlying assumption that any allele can mutate into any other allele from the allelic set. However, for a genetic sequence, this is not always true. To describe situations where the possible mutational transitions are constrained, we introduce the allele graph $\mathcal{A}$ [10]. This graph is defined over the allelic set and has an adjacency matrix $A = \{A_{\mu\nu}\}_{\mu,\nu=0,\dots,a-1}$ with

$$A_{\mu\nu} = \begin{cases} 1, & \text{if the transition from } \mu \text{ to } \nu \text{ is possible}, \\ 0, & \text{otherwise}. \end{cases} \tag{5}$$

The sequence space can then be written as the $L$-fold Cartesian product $\mathcal{A}^L$ and the Hamming graph $\mathbb{H}_a^L$ is the special case when the allele graph $\mathcal{A}$ is complete. Some examples of allele graphs are depicted in fig. 2.

- **Fitness landscape.** From an evolutionary perspective, **fitness** is a measure of the reproductive capacity of an organism and is typically quantified by the mean number of offspring. A fitness landscape is a map $g : \mathcal{A}^L \rightarrow \mathbb{R}$ that assigns a fitness value to each genotype sequence. In the following we will assume that $g$ is non-degenerate, i.e., no two genotypes have the same fitness value.

- **Fitness graph.** By orienting the links between mutational neighbors in the direction of increasing fitness, a fitness landscape $g$ naturally induces an orientation on the genotype graph $\mathcal{A}^L$ which by construction is acyclic. The resulting acyclic oriented graph is referred to as a fitness graph [11–15]. Given a fitness graph, one is typically interested in the following two features:

  1. **Peaks.** These are local or global maxima of the fitness landscape. Peaks are the culminating points (targets) of natural selection and appear as sinks in the fitness graph.
  2. **Accessible paths.** These are paths (sequences of connected genotypes) on the fitness landscape with monotonically increasing fitness. In other words, an accessible path respects the orientation of the fitness graph. This definition of accessibility is similar to the concept of reachability introduced in [16].

## 3 Wright's rugged vs. Fisher's accessible landscapes

The study of fitness landscapes dates back to the first half of the $20^{th}$ century. Since the beginning, these studies have been motivated by a quest to understand the predictability and repeatability of evolution [4]. At the core of this quest, one inevitably finds a debate of ruggedness vs. accessibility. Sewall Wright, who introduced fitness landscapes in 1932, was of the opinion that these are highly *rugged* which implies that it is rather difficult for an evolving population to reach the global fitness maximum because of the many local maxima present in the landscape [17]. Interestingly, Ronald Fisher, another foundational personality in the field of evolutionary theory, had a contradictory opinion [18]. He argued that the higher the number of dimensions, the greater the number of inequalities that must be satisfied for a genotype to be a fitness maximum, as opposed to a single inequality in one dimension. Consequently, most genotypes would turn out to be saddles rather than peaks.

To what extent real fitness landscapes are rugged or accessible is, at heart, an empirical question which has begun to be explored over the past two decades [4, 5, 11, 19–23]. A fully consistent picture is still to emerge from this work. For example, a systematic study of 15 point mutations required to mutate the spike protein responsible for cell entry from the Wuhan strain of SARS-CoV2 to the Omicron BA.1 variant found that the latter was not accessible via any of the 15! direct paths,[1] since intermediate genotypes tended to lead to a lower affinity of the protein to the cell wall receptor [24]. The interpretation offered by the authors of this study is that the Omicron strain spread in the population not because of higher transmissability, but because of its ability to overcome the human immune defense; stated differently, receptor affinity is an inappropriate fitness proxy in this case. On the other hand, a recent study inspecting nine nucleotide positions of the DHFR gene in *Escherichia coli* under antibiotic selection pressure revealed that the resulting landscape is highly rugged but also highly accessible [25] (see also [26]). To eludicate the relation between ruggedness and accessibility in fitness landscapes is a major goal of the theoretical investigations that will be described in these notes [27].

---

[1]See Sect. 4.2 for a definition of this term.

# 4 House of Cards model

## 4.1 Ruggedness and complexity

The House of Cards (HoC) landscape is a model where the fitness values assigned to the elements of the sequence space, $g(\sigma)$, are i.i.d. continuous random variables [28,29]. We first try to gauge the level of ruggedness present in this basic fitness model. The question we therefore ask is the following: under this model, what is the probability that a certain genotype $\sigma$ is a peak?

To answer this, consider a set of genotypes in a sequence space $\mathbb{H}_a^L$ where $a$ and $L$ are as defined before. For a sequence to be an immediate neighbour of a given reference sequence in this set, it should have a Hamming distance of one from the latter. This means that a neighbour will have any one out of all $L$ positions differing from the reference and this differing position can have any one of the remaining $a-1$ values to choose from. So, a given sequence in this space will have $(a-1)L$ immediate neighbours. We pick a random sequence $\sigma^{(0)}$ with fitness $g_0$ which has neighbours $\{\sigma^{(i)}\}$ with corresponding fitnesses denoted by $\{g_i\}$ where $i \in \{1, ..., (a-1)L\}$.

We define $\mathcal{P}_{max}$ as the probability that a random genotype $g_0$ is a peak, i.e., it has the highest fitness among all its immediate neighbours. By symmetry [29]

$$\mathcal{P}_{max} = \text{Prob}\left[g_0 = \max\left\{g_0, g_1, g_2, ..., g_{(a-1)L}\right\}\right] = \frac{1}{(a-1)L+1}. \tag{6}$$

We next define the random variable $N_L$ as the number of peaks of the landscape. The expected number of peaks is then given by

$$\mathbb{E}(N_L) = \mathcal{P}_{max}a^L = \frac{a^L}{(a-1)L+1}. \tag{7}$$

In a sense, this simple result already suffices to resolve the controversy between Sewall Wright and Ronald Fisher described in Section 3 [30]. Equation (6) tells us that $\lim_{L\to\infty} \mathcal{P}_{max} = 0$ which supports Fisher's argument: A local maximum is a rare occurrence in a high-dimensional landscape. On the other hand, (7) tells us that $\mathbb{E}(N_L) \to \infty$ for $L \to \infty$ in conjunction with Wright's argument that such landscapes typically possess a large number of peaks.

One can also find the variance of the number of peaks to be [31]

$$Var(N_L) = \frac{a^L(a-1)(L-1)}{2\left\{(a-1)L+1\right\}^2}. \tag{8}$$

In the limit $L \to \infty$

$$Var(N_L) \to \frac{1}{2}\frac{a^L}{(a-1)L} \approx \frac{1}{2}\mathbb{E}(N_L), \tag{9}$$

which shows that the number of peaks has sub-Poissonian statistics: Because two neighboring genotype sequences cannot both be peaks, there is a local repulsion between peak genotypes which reduces the fluctuations compared to a completely random distribution.

The exponential dependence of eq. (7) on $L$ suggests to define the complexity $\Lambda$ of the landscape as the exponential growth rate of the expected number of peaks with the dimensionality [1]

$$\Lambda = \lim_{L\to\infty} \frac{1}{L} \log \mathbb{E}(N_L). \tag{10}$$

From eq. (7), $\Lambda = \ln a$, which means that to exponential order the number of peaks is comparable to the number of nodes in the graph, i.e., the landscape is (in this sense) maximally rugged.

Note that in (10) we have defined $\Lambda$ in terms of the logarithm of the expectation $\log \mathbb{E}(N_L)$. The resulting quantity is known as the *annealed* complexity [1]. For random landscapes in which the number of peaks fluctuates strongly from one realization to the other, the annealed complexity may not coincide with its *quenched* counterpart defined in terms of $\mathbb{E}(\log N_L)$ [9, 32]. For the HoC model the two definitions coincide, because [as can be read off from (7) and (8)] the relative fluctuations in $N_L$ vanish for $L \to \infty$ and the number of peaks satisfies a central limit theorem [33].

## 4.2 Accessibility

Consider two genotypes $\alpha, \omega \in \mathcal{A}^L$ and a path of length $l$ connecting them. Let $\{\sigma^{(i)}\}$ be the set of sequences that this path goes through with the index $i$ representing the order in which they appear in the path with $i \in \{0, 1, \ldots, l\}$. The path can be represented as

$$\alpha = \sigma^{(0)} \to \sigma^{(1)} \to \sigma^{(2)} \to \ldots \to \sigma^{(l)} = \omega,$$

such that any two consecutive sequences in the path are at a unit distance to each other,

$$d(\sigma^{(i)}, \sigma^{(i+1)}) = 1. \tag{11}$$

The path is called *accessible* if fitness always increases monotonically along the path [19, 20, 30, 34],

$$g(\sigma^{(i+1)}) > g(\sigma^{(i)}). \tag{12}$$

It is *direct* if the distance from the first sequence $\alpha$ in the path also increases monotonically along the path to the final sequence. Given eq. (11), this implies that the distance, of any sequence in a direct path, from the starting point is given by its position in the path,

$$d(\alpha = \sigma^{(0)}, \sigma^{(i)}) = i. \tag{13}$$

Otherwise, the path is *indirect*. Figure 3 shows examples of direct and indirect paths between two genotypes on a three dimensional binary Hamming graph.

In the following, we take a closer look at the problem of accessibility in the HoC landscape. First, we restrict ourselves to direct paths, and inspect the probability of existence of at least one such path conditioned on the fitness gain between the initial and the final sequences. We then expand our radar to include the indirect paths and expound on how the resulting dramatic increase in accessibility is tackled.

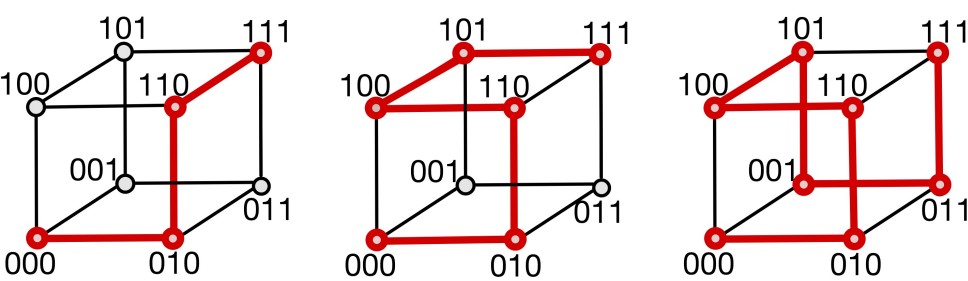

Figure 3: **Direct and indirect paths connecting the corners $\alpha = 000$ and $\omega = 111$ of the binary 3-cube.** The figure shows a direct path of length $l = d(\alpha, \omega) = 3$ to the left and two indirect paths of length $l = d + 2 = 5$ and $l = d + 4 = 7$ next to it. The rightmost path visits all nodes and is the longest possible self-avoiding path.

### 4.2.1 Accessibility via direct paths

Note that the accessibility properties of the HoC model, as well as the results on fitness maxima discussed in Sect. 4.1, rely only on the rank order of fitness values and are therefore manifestly independent of the fitness distribution (provided it is continuous). To simplify the following considerations we will nevertheless specify the fitness values to be uniformly distributed on the interval $[0, 1]$.

By definition eq. (13), all direct paths between the given genotypes $\alpha$ and $\omega$ have a length $l = d(\alpha, \omega)$. This implies that the two sequences differ at $l$ sites and $l!$ direct paths can be constructed between them. Given a fitness difference of $\beta \in (0, 1]$ between the two sequences such that $g(\omega) = g(\alpha) + \beta$, a direct accessible path would pass through $l - 1$ intermediate sequences. The fitness values of the intermediate sequences should lie in $(g(\alpha), g(\alpha) + \beta)$ which is true with probability $\beta^{(l-1)}$, and additionally they should be traversed in order of increasing fitness. There are $(l-1)!$ equiprobable orderings of the $l-1$ i.i.d. random variables $\{g(\omega^{(i)})\}_{i=1,\dots,l-1}$, and the probability that they display the correct ordering is $1/(l-1)!$. Therefore, the probability for a direct path between two sequences $\alpha$ and $\omega$ that are at distance $l$ and differ in fitness by $\beta$ to be accessible can be written as

$$\mathcal{P}_{\beta,l} = \frac{\beta^{l-1}}{(l-1)!} \,. \tag{14}$$

With this knowledge at hand, the expected value of the number $X_{\alpha,\omega}$ of such paths is simply the product of the number of direct paths and eq. (14),

$$\mathbb{E}(X_{\alpha,\omega}) = l! \frac{\beta^{l-1}}{(l-1)!} = l\beta^{l-1} \,. \tag{15}$$

In the limit $l \to \infty$, this quantity vanishes as long as $\beta < 1$. Markov's inequality further bounds the probability of having at least one direct path by the mean number of direct paths as

$$\mathcal{P}[X_{\alpha,\omega} \geq 1] = \sum_{k=1}^{\infty} \mathcal{P}[X_{\alpha,\omega} = k] \leq \sum_{k=1}^{\infty} k\mathcal{P}[X_{\alpha,\omega} = k] = \mathbb{E}(X_{\alpha,\omega}), \tag{16}$$

and therefore also the former quantity vanishes for sequences that are very far apart on the Hamming graph unless the fitness difference between them is exactly 1. When $\beta = 1$ the bound imposed by eq. (16) on $\mathcal{P}[X_{\alpha,\omega} \geq 1]$ diverges and is not informative.

A more careful analysis extending to the second moment of $X_{\alpha,\omega}$ reveals a threshold based increase in $\mathcal{P}[X_{\alpha,\omega} \geq 1]$ with a threshold fitness difference $\beta_c$ lying close to one. To be precise, defining an $l$-dependent threshold $\beta_c(l)$ as

$$\beta_c(l) = 1 - \frac{\ln l}{l} \,, \tag{17}$$

Hegarty and Martinsson [35] proved that the probability of existence of an accessible paths displays a percolation-like phase transition as a function of the fitness difference between the initial and final genotype in the following sense:

$$\lim_{l \to \infty} \mathcal{P}[X_{\alpha,\omega} \geq 1 | \beta = \beta_l] = \begin{cases} 0, & \text{for } \beta_l < \beta_c(l), \\ 1, & \text{for } \beta_l > \beta_c(l). \end{cases} \tag{18}$$

As we will see in the following, also in more general settings the onset of accessibility often occurs when the expected number of accessible paths $\mathbb{E}(X_{\alpha,\omega})$ is equal to unity. Correspondingly, one can arrive at eq. (17) by solving

$$\mathbb{E}[X_{\alpha,\omega} | \beta = \beta_c(l)] = 1 \,, \tag{19}$$

for large $l$.

### 4.2.2 Accessibility via indirect paths

Once the imposition to be direct is removed, accessibility increases significantly. This may seem counter-intuitive considering that indirect paths are longer and therefore have a lower probability to be accessible. However, this is surpassed by the sheer increase in the number of possible paths between two sequences on removal of this constraint [36, 37].

Let us first consider the case of the binary hypercube (fig. 3). For two sequences at distance $d$, the length $l$ of an indirect path satisfies $l = d + 2p$, where $p$ is the number of back-steps i.e. reversals of a mutation $(1 \rightarrow 0)$. The total excess length is $2p$, because each reversal has to be compensated by an additional forward step $(0 \rightarrow 1)$. Additionally, the path must satisfy the fundamental constraint of acyclicity, i.e. it must be self-avoiding. The longest self-avoiding path passes through all nodes of the graph and has length $2^L - 1$. For $a > 2$ there are not only back-steps but also "sideways" steps that lead to an allele that is present neither in the initial nor in the final sequence of the path [37, 38].

Unlike in the analysis of direct paths, one cannot simply use the knowledge of the number of indirect paths between $\alpha$ and $\omega$ because estimating it becomes increasingly difficult for larger $L$ [39]. Schmiegelt and Krug [10] came up with a clever solution to this problem by introducing an extended fitness landscape where each node on a fitness graph (genotype) is assigned a countably infinite sequence of i.i.d. fitness values. A self-intersecting path on the original landscape can then be mapped to a self-avoiding path by invoking the $k^{th}$ subsequent fitness value from the sequence of fitnesses at the $k^{th}$ recurrence of a node in the path. If such a path on the extended landscape is accessible, the corresponding path on the original landscape is called 'quasi-accessible'. Schmiegelt and Krug [10] proved that the probability of having at least one quasi-accessible path is equal to that of having at least one accessible path. Therefore, the problem can be modified to look for quasi-accessibility and the difficulty associated with enumerating all self-avoiding paths is bypassed by expanding the analysis to all possible (generally intersecting) paths.

It is shown in [10] that the expected number of quasi accessible paths $\mathbb{E}[\tilde{X}_{\alpha,\omega}]$ between genotypes $\alpha$ and $\omega$ with a fitness difference $\beta$ depends exponentially on $L$ as

$$\mathbb{E}[\tilde{X}_{\alpha,\omega}] \sim \prod_{k,l=0}^{a-1} \left[ (e^{\beta A})_{k,l} \right]^{p_{k,l} L} . \tag{20}$$

Here, $p_{k,l}$ is the fraction of sites where $\alpha_i = k$ and $\omega_i = l$, and $A$ is the adjacency matrix of the allele graph. The matrix $p_{k,l}$ encodes the scaled distance $\delta$ between the endpoints through

$$\delta = \lim_{L \to \infty} \frac{1}{L} d(\alpha, \omega) = 1 - \sum_{k=0}^{a-1} p_{kk} . \tag{21}$$

Guided by the analysis of the directed case in the preceding section, we consider the condition $\lim_{L \to \infty} \mathbb{E}[\tilde{X}_{\alpha,\omega}] = 1$. By eq. (20) this translates into

$$\mathbb{E}[\tilde{X}_{\alpha,\omega}]^{\frac{1}{L}} = \prod_{k,l=0}^{a-1} \left[ (e^{\beta A})_{k,l} \right]^{p_{k,l}} = 1 . \tag{22}$$

It is easy to see that the left hand side of this equation is an increasing function of $\beta$. Denoting the solution of eq. (22) by $\beta^*$, it then follows from Markov's inequality (16) that $\lim_{L \to \infty} \mathcal{P}[X_{\alpha,\omega} \geq 1] = 0$, i.e., accessible paths do not exist, for $\beta < \beta^*$. This shows that $\beta^*$ provides a lower bound on the accessibility threshold $\beta_c$. In particular, since $\beta \in (0, 1]$ by construction, there are no accessible paths at any fitness difference if $\beta^* > 1$. Moreover, for a large class of allele graphs (including in particular the complete graph over $a$ nodes) it can be shown that the lower bound $\beta^*$ is tight and the accessibility threshold is given exactly by the

solution of eq. (22) [10]. If the genotype space is the binary Hamming graph, the condition takes the simpler form [39]

$$\sinh(\beta_c)^\delta \cosh(\beta_c)^{1-\delta} = 1. \tag{23}$$

## 5 Structured landscapes

Though plenty of interesting properties underscore their importance as an attractive playfield, completely random landscapes discussed so far are not realistic because the idea that every single mutation replaces an organisms fitness by an independent random variable is bizarre [28]. This motivates the study of structured landscape models where fitness values of nearby sequences are correlated.

Here we briefly mention some paradigmatic examples.

- **Kauffman's NK model**: The NK model [40–42] imposes structure on the landscape by introducing interaction sets $\mathcal{B}_i \subseteq \mathcal{L} \equiv \{1, \dots, L\}$, where $i = 1, \dots, b$ and the size of the subsets is $|\mathcal{B}_i| = k \leq L$. The fitness effect of a mutation is affected by the loci in the subsets that it belongs to, but independent of other loci. This is achieved by writing the fitness of a sequence $\sigma$ as a sum of contributions

$$g(\sigma) = \sum_{i=1}^{b} g_i(\sigma_{i,1}, \sigma_{i,2}, ..., \sigma_{i,k}), \tag{24}$$

  where the index $i, j$ denotes the $j$'th element of the set $\mathcal{B}_i$ and $g_i$ is a function that assigns an independent random variable to each of its $a^k$ arguments. Thus, each $g_i$ is an independent HoC fitness landscape on the Hamming graph $\mathbb{H}_a^k$.

  The subsets $\mathcal{B}_i$ can be tailored to suit the underlying biological context. They can be systematically overlapping, disjoint or completely random. By tuning the parameter $k$ one can effectively control the degree of ruggedness in the landscape. For $k = 1$ the alleles contribute independently to fitness and can be individually optimized, which means that the landscape is single-peaked. On the other hand, for $k = L$ the $g_i$ are random functions defined on the entire sequence space $\mathbb{H}_a^L$ and the HoC model is recovered.

  Mathematical results for the NK model have been obtained primarily in the case of binary alleles ($a = 2$), and we provide a brief summary in the following. A key insight is that the landscape properties in the large $L$ limit differ qualitatively depending on whether this limit is carried out at fixed $k$, or as a joint limit $k, L \to \infty$ at fixed ratio $k/L < 1$ [42]. In the latter case NK landscapes are similar to HoC landscapes, in the sense that the exponential complexity (10) takes on its maximal value $\Lambda = \ln 2$. For fixed $k$ the complexity is less than $\ln 2$ and generally depends on both $k$ and the choice of interaction sets. Remarkably, while fixed $k$ NK landscapes are less rugged than HoC landscapes, they are much less accessible: For most commoly used choices of interaction sets, the existence of accessible paths between genotypes at distance $d(\alpha, \omega) \sim L$ is exponentially unlikely for large $L$ [27, 42].

- **Rough Mount Fuji model**: The RMF model [43, 44] defines the fitness of a sequence as a function of the distance from a reference, $\sigma^\star$ (typically a high fitness genotype) with some linearly added stochasticity,

$$g(\sigma) = -cd(\sigma, \sigma^\star) + \xi_\sigma, \tag{25}$$

  where $c > 0$ is a constant and $\xi_\sigma$ are i.i.d. continuous random variables.

The degree of randomness in the landscape can be varied by modifying $c$. For $c = 0$, the resulting landscape is random and for sufficiently large $c$ (larger than the variability of the $\xi_\sigma$) one gets a smooth (Mt. Fuji-like) landscape with a single global fitness peak. Explicit expressions for the expected number of peaks can be derived for specific choices of the distribution of $\xi_\sigma$ [44]. Moreover, Hegarty and Martinsson have proved that direct accesible paths to the reference sequence $\sigma^\star$ exist with probability 1 for $L \to \infty$ whenever $c > 0$ [35].

- **Landscapes with an intermediate phenotype**: These landscapes are generated using a fitness function defined as a composition of a linear and a non-linear map on the sequence space. In the simplest case the linear part maps the sequence to a scalar real number $z(\sigma)$ and the composition takes the form [45]

$$g(\sigma) = \Phi\left[z(\sigma)\right], \qquad z(\sigma) = \sum_{i=1}^{L} \sum_{\mu=0}^{a-1} a_{i,\mu} \delta_{\sigma_i, \mu}, \tag{26}$$

where the $a_{i,\mu} \in \mathbb{R}$ are random coefficients representing the effect of allele $\mu$ at site $i$, and $\Phi$ is a non-linear function.

One motivation for such a composition is to capture, to some extent, the complex biological interactions that determine the fitness of an individual. Organismal fitness is a consequence of the interaction of an organism with its environment. However, the genotype of an organism does not directly interact with the environment. Rather, the genotype governs the physical characteristics of the organism, referred to collectively as its *phenotype*, which then in turn interacts with the environment. The latter interaction determines the reproductive success of the organism which, through the heredity of the genetic information, modifies the genotypic composition of the population. A schematic of this feedback cycle is depicted in fig. 4.

In eq. (26), the linear combination $z(\sigma)$ represents the genotype-phenotype map and $\Phi$ represents the phenotype-fitness map. The scalar phenotype $z$ is maximized by the unique sequence $\sigma^*$ obtained by setting each $\sigma_i^*$ to the allele with the largest coefficient $a_{i,\mu}$. If the map $\Phi$ is monotonic, this property is inherited by the fitness landscape $g$ which is then single-peaked. Multi-peaked landscapes can emerge if $\Phi$ is non-monotonic, which implies that an intermediate value of the phenotype is favored by selection [46], or if the phenotype is multi-dimensional [21, 45].

The best known example of a composite genotype-phenotype-fitness map is Fisher's geometric model (FGM) that was introduced almost a century ago with the aim to elucidate the effect of organismal complexity on the adaptive process [7], see [47] and [8, 9] for a modern perspective. In the standard setting of FGM the phenotype is represented by a real valued vector $\vec{z} \in \mathbb{R}^n$, where the phenotypic dimension $n$ quantifies the organismal complexity, and the phenotype-fitness map $\Phi(\vec{z})$ is a radially symmetric function with a unique optimum at the origin (see Sect. 7 for further discussion of this model).

For the remaining sections, we will focus on composite landscapes, mostly with a one-dimensional phenotype as defined in eq. (26). Moreover, although many of the results that we will describe can be generalized to the multiallelic setting, for convenience we consider only binary genotype sequences ($a = 2$).

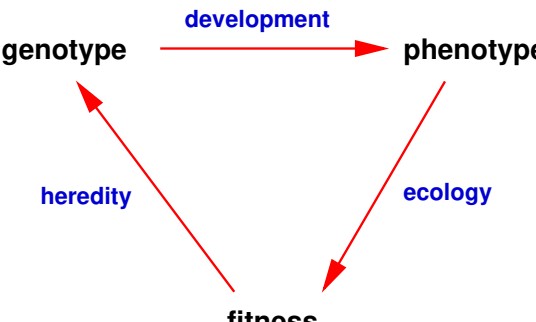

Figure 4: **Reductionist biology.** The feedback cycle of the key elements of Darwinian evolution. The genotype encodes all the ingredients for the development of an organism characterised by its phenotype (its physical features). Fitness, i.e. the reproductive success of the organism, is determined by its ecological interactions with the (biotic and abiotic) constituents of its environment. Based on its reproductive success, the relative abundance of its genotype in the next generation is determined through the heredity of the genetic information. The figure was inspired by a lecture by Amitabh Joshi at JNCASR Bangalore.

## 6 Epistasis and submodularity

### 6.1 The concept of epistasis

In classical genetics, epistasis is defined as the phenomenon that the effect of one gene variant is cancelled by the presence of another [48]. As a simple example, consider the effect of a gene variant causing baldness on the gene responsible for hair colour. In the presence of the former, the effect of the latter would be completely masked. In modern usage, the term refers broadly to any kind of interaction between mutational effects, both between and within genes [5, 34, 49, 50]. A conceptual difficulty with this definition is that, in order to identify interactions between mutational effects, one needs to specify a non-interacting baseline [51].

Consider first the simplest possible setting of binary sequences of length $L = 2$, and let the sequence $\sigma = 00$ represent the reference genotype or 'wild type'. With respect to it, the sequence 11 has two mutations and is therefore called 'double mutant'. We denote by $s_{1/2}$ the fitness change in mutating the first(/second) site of the wild type, i.e.

$$s_1 = g(10) - g(00), \tag{27}$$
$$s_2 = g(01) - g(00). \tag{28}$$

If we assume no interactions between these two mutations, the expected fitness of the double mutant, under an *additive* null model, would be given by

$$g_{add}(11) = g(00) + s_1 + s_2 = g(10) + g(01) - g(00). \tag{29}$$

However due to epistasis, the actual fitness of the double mutant might differ from this prediction. The pairwise epistatic interaction $\epsilon_{12}$ between the two mutants can therefore be quantified by the deviation from the additive expectation,

$$\epsilon_{12} = g(11) - g_{add}(11) = g(11) + g(00) - g(10) - g(01). \tag{30}$$

This reasoning can be generalized to larger $L$ [13, 49, 52, 53]. For example, for $L = 3$ there are three pairwise epistatic interactions correponding to the double mutants 110, 101 and 011 and additionally a third order interaction

$$\epsilon_{123} = g(000) + g(110) + g(101) + g(011) - g(100) - g(010) - g(001) - g(111). \tag{31}$$

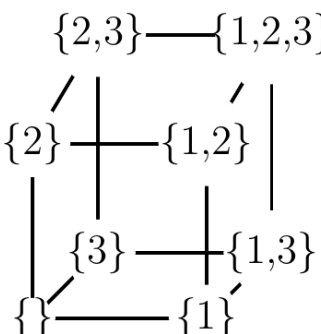

Figure 5: **Hasse diagram.** The diagram represents the nodes of the binary hypercube in three dimensions as elements of the power set $\mathcal{P}\{1,2,3\}$. Courtesy of Daniel Oros.

For binary genotypes of length $L$, there are $2^L - (L+1)$ epistatic interactions. Together with the $L$ selection coefficients $s_i$ describing the effects of single mutations and the wild type fitness $g(0\ldots0)$, the number of coefficients is thus equal to the number of genotypes. This shows that the decomposition into epistatic interactions retains the full information about the fitness landscape.

## 6.2 Universal epistasis and submodularity

For the purposes of the following considerations, it is useful to write genotypes in set notation [15, 54]. This is done by noting that the binary hypercube $\{0,1\}^L$ is isomorphic to the power set $\mathcal{P}(\mathcal{L}), \mathcal{L} = \{1,..,L\}$, such that for any $\sigma = (\sigma_1, \sigma_2, ..., \sigma_L) \in \{0,1\}^L$,

$$\sigma \to \{i : \sigma_i = 1\} \in \mathcal{P}(\mathcal{L}). \tag{32}$$

This mapping is illustrated in Figure 5. In the set representation $\sigma$ comprises the subset of sites which have been mutated relative to the wild type sequence $0\ldots0$.

Using the set representation, we can formalize a global constraint on the fitness landscape that was first described in [15]. Consider two genotypes $\sigma, \sigma'$, where $\sigma'$ is a subset of $\sigma$, as well as a set $\tau$ of mutations that are present in neither $\sigma$ nor $\sigma'$, i.e., $\tau \subseteq \mathcal{L} \backslash \sigma$. Then we say that the fitness landscape displays *universal epistasis* if the fitness effect of adding the mutations in $\tau$ is always smaller (or always larger) in the genetic background of $\sigma$ than in the background $\sigma'$. Specifically, the condition of universal negative epistasis (UNE) reads [27]

$$g(\sigma \cup \tau) - g(\sigma) \leq g(\sigma' \cup \tau) - g(\sigma'). \tag{33}$$

Universal epistasis implies that the partial order induced by the subset-superset relation between genotypes is inherited by the fitness effect of mutations on the background of these genotypes.

Equation (33) turns out to be equivalent to a property known in discrete mathematics as *submodularity* for set functions [27]. On defining $\sigma = A$ and $\sigma' \cup \tau = B$, it follows that $\sigma \cup \tau = A \cup B$ and $\sigma' = A \cap B$. Using these definitions and re-writing eq. (33), the inequality takes on the standard form of the condition for submodularity [55, 56]

$$g(A \cup B) + g(A \cap B) \leq g(A) + g(B), \qquad \forall A, B \in \mathcal{P}(\mathcal{L}). \tag{34}$$

The equivalence of Eqs. (33) and (34) was first pointed out in [57]. Submodularity is a key concept in discrete combinatorial optimization, because it is a property that arises naturally in many applications. The maximization of submodular set functions is an NP-hard problem,

but general theorems exist that provide bounds on the quality of approximation algorithms, see e.g. [58].

Conversely, one can define *universal positive epistasis* [15] by simply flipping the inequality in eq. (33) and show that it corresponds to the property of *supermodularity* written by similarly flipping the inequality in eq. (34). In the following, we will keep the discussion focused to UNE because of its important consequences for the accessibility of fitness peaks (see Sect. 7). By symmetry, universal positive epistasis and supermodularity have analogous consequences for the accessibility of the local minima of the fitness function via fitness-monotonic paths.

## 6.3 Construction of submodular landscapes

The concept of universal epistasis was originally formulated in the context of a geometric classification of fitness landscapes that is based on the triangulation that a landscape induces in the *continuous* cube $[0,1]^L$, see [59] for details. Specifically, Crona et al. [15] showed that landscapes that induce a particular (staircase) triangulation satisfy the condition of universal positive epistasis. While they also reported a certain enrichment of this property in an empirical data set, it is currently unknown to what extent biological fitness landscapes should be expected to satisfy conditions of universal epistasis, sub- or supermodularity. Here we show, for a simple example, how submodularity may arise from a composite genotype-phenotype-fitness map of the type discussed in Sect. 5.

To see this, we first recall the defining property of a concave function $\Phi : \mathbb{R} \to \mathbb{R}$

$$\Phi(x+z) - \Phi(x) \le \Phi(y+z) - \Phi(y), \qquad \forall x > y, \quad z > 0, \tag{35}$$

and note its similarity to the UNE condition (33). Indeed, it is often stated that sub- and supermodularity can be regarded as a generalization of concavity or convexity to set functions [56,57]. Consider now a fitness landscape of the form (26) where we assume binary genotypes ($a = 2$) and moreover impose the condition that the coefficients $a_i$ of the linear genotype-phenotype map are positive, $a_i > 0$. Then, by construction, the phenotype

$$z(\sigma) = \sum_{i=1}^{L} a_i \sigma_i, \tag{36}$$

is a monotonically increasing function of the number of mutations in a genotype and therefore $z(\sigma') < z(\sigma)$ for any $\sigma' \subset \sigma$. If we further assume that $\Phi$ is concave, then

$$\begin{aligned} g(\sigma \cup \tau) - g(\sigma) &= \Phi[z(\sigma) + z(\tau)] - \Phi[z(\sigma)] \\ &\le \Phi[z(\sigma') + z(\tau)] - \Phi[z(\sigma')] = g(\sigma' \cup \tau) - g(\sigma'), \end{aligned} \tag{37}$$

where the inequality follows from the concavity property (35), and $z(\sigma \cup \tau) = z(\sigma) + z(\tau)$ because $\sigma \cap \tau = \emptyset$. This is precisely the condition (33) of submodularity or UNE. With a little more work, one can verify that submodularity is obeyed (in suitably transformed sequence space coordinates) even when the positivity condition on the $a_i$s is relaxed [27].

In the present context our main interest is in submodular fitness landscapes that have multiple peaks. This can be achieved by choosing $\Phi$ to be non-monotonic [45], see fig. 6 for illustration. With this choice the model becomes a version of Fisher's geometric model (FGM), see Sect. 5, and we conclude that the fitness landscape of FGM with a one-dimensional phenotype is submodular (satisfies the condition of UNE) if the phenotype-fitness map $\Phi$ is concave.

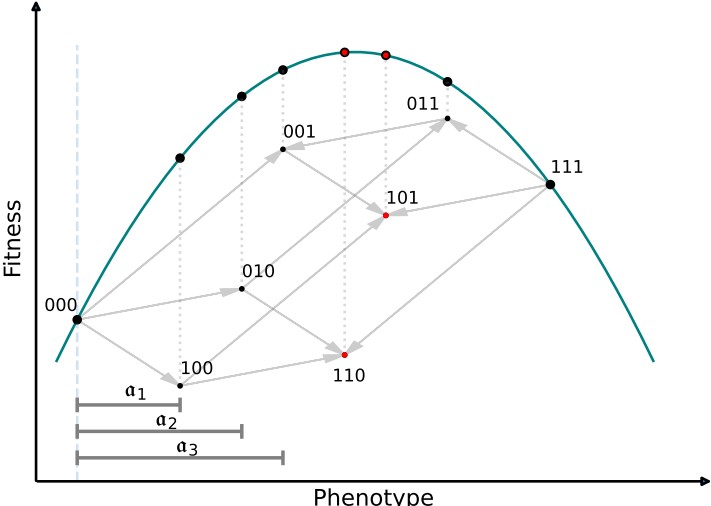

Figure 6: **Submodular landscapes construction.** A submodular landscape constructed by convolution of a linear genotype-phenotype map and a concave phenotype-fitness map. The three individual mutations increase the phenotypic value by different amounts, and the phenotypes of all other genotypes are linear combinations of these effects. Then, a concave function maps the phenotypes to their fitness values. The non-monotonicity of this function leads to a fitness graph with multiple peak (two in this case – marked in red). The fitness graph in faint grey illustrates the rank ordering of the fitness values.

The problem of maximising fitness in fitness landscapes is analogous to minimizing energy in a physical system, and correspondingly the concept of submodularity can be applied to disordered spin models as well. As an example, we choose the (concave) phenotype-fitness map $\Phi = -z^2$ in Fisher's model and consider the Hamiltonian

$$H = -\Phi = z^2. \tag{38}$$

With the appropriate change of the $\sigma_i$'s to spin variables $\eta_i = 1 - 2\sigma_i$, one arrives at the Hamiltonian of an antiferromagnetic Hopfield model with continuous patterns and random fields [9, 60],

$$H = \sum_{i,j} J_{ij} \eta_i \eta_j + \sum_i h_i \eta_i, \tag{39}$$

with

$$J_{ij} = \frac{1}{4} a_i a_j, \tag{40}$$

and

$$h_i = -\frac{1}{2}\left(\sum_j a_j\right) a_i. \tag{41}$$

Based on the previous discussion, it follows that the Hamiltonian defined by Equations (39-41) is a supermodular function. Note that, by construction, the ground state of the model is $\eta_i \equiv 1$ ($\sigma_i \equiv 0$).

# 7   Accessibility of submodular landscapes

In this section, the reward for establishing the submodularity property can finally be reaped with respect to one of the broader themes of this text, i.e., the accessibility of local fitness peaks via fitness-increasing paths. We already demonstrated that Fisher's geometric model generates rugged submodular fitness landscapes when certain conditions are met. Here, we show that a specific kind of accessibility property is implicit to the construction of submodular landscapes.

## 7.1   The accessibility property

The aforementioned property is known by the name of *subset-superset accessibility property* (AP), and was first identified for a landscape model with two intermediate phenotypes designed to describe the evolution of antibiotic resistance [54, 61, 62]. We say that a landscape has the AP if **any peak is accessible from all its sub- and supersets along all direct paths**. It is trivial to see that any peak in such a landscape would always be accessible from the zero-string i.e. $\sigma = \varnothing$ and the one-string $\sigma = \mathcal{L}$.

To prove that submodular landscapes have the AP, we consider a peak genotype $\sigma$. This means that any immediate neighbours in the subset or superset of $\sigma$ would have a fitness lower than $\sigma$ i.e.,

$$g(\sigma \cup \{i\}) - g(\sigma) < 0, \tag{42}$$

and

$$g(\sigma) - g(\sigma \setminus \{j\}) > 0, \tag{43}$$

for all $j \in \sigma$, $i \in \mathcal{L} \setminus \sigma$. Now consider a subset genotype $\sigma' \subseteq \sigma$ and a mutation $k \in \sigma \setminus \sigma' \subset \sigma$. Then by the UNE condition (33), we have

$$g(\sigma' \cup \{k\}) - g(\sigma') \geq g(\sigma) - g(\sigma \setminus \{k\}) > 0. \tag{44}$$

This implies that the mutation $k$ is fitness increasing on the background $\sigma'$, ergo the corresponding step on the direct path from $\sigma'$ to $\sigma$ is accessible. One can similarly prove the accessibility from a genotype in the superset of $\sigma$.

The proof outlined above was first presented in [27], but in fact the AP of submodular landscapes has been known for a long time in the Russian literature on combinatorial optimization, where it is attributed to a 1962 article by Viktor Pavlovich Cherenin [55, 57]. It was rediscovered in 1974 by Frieze [63].

Importantly, while submodularity implies the AP, the two properties are not equivalent. Indeed, the AP uses only the rank ordering of fitness values and, as such, it is invariant under arbitrary monotonic transformations of the fitness values. This is in contrast to the submodularity conditions (33, 34) which depend on the actual values taken by the function $g$. As an application of this observation, consider the example of FGM with a one-dimensional phenotype discussed in Sect. 6.3. Whereas submodularity is guaranteed for this model only when the phenoype-fitness map $\Phi$ is concave, the AP holds much more generally, for any function $\Phi$ that has at most one maximum, that is, for which the derivative changes sign at most once. This includes in particular the Gaussian fitness function $\Phi(z) = e^{-z^2}$ that is often assumed in the context of FGM [8, 9, 47], as well as more generally in the description of the evolution of phenotypic traits [64, 65].

Exploiting the fact that a positive linear combination of submodular functions is also submodular [56], we can further establish the AP for a special case of FGM with a multidimensional phenotype. Consider the phenotype-fitness map

$$\Phi(\vec{z}) = \exp\left[-|\vec{z}|^2\right] = \prod_{k=1}^{n} e^{-z_k^2}, \tag{45}$$

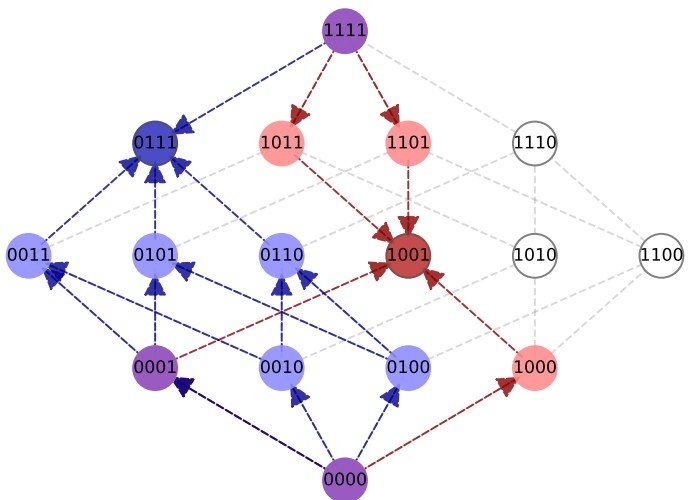

Figure 7: **Illustration of the subset-superset accessibility property.** Two peak genotypes (shown in dark red and dark blue) in the fitness graph are accessible from all their sub- and supersets (depicted in light red and light blue) as indicated by the arrows.

and let the components $z_k$ of the $n$-dimensional phenotype vector $\vec{z}$ depend on the genotype through the linear relation

$$z_k(\sigma) = q_k + \sum_{i=1}^{L} a_{i,k} \sigma_i, \quad k = 1, \ldots, n, \tag{46}$$

where the $q_k$ are arbitrary constants and the coefficients $a_{i,k} > 0$. Then the logarithm of the fitness function $g(\sigma) = \Phi[\vec{z}(\sigma)]$

$$\ln g(\sigma) = \sum_{k=1}^{n} - \left( q_k + \sum_{i=1}^{L} a_{i,k} \sigma_i \right)^2, \tag{47}$$

is a sum of concave functions of a linear genotype-phenotype map with positive coefficients. On the basis of the arguments of Sect. 6.3 we conclude that $\ln g$ is submodular, and therefore $g$ has the AP. Note that, in contrast to the one-dimensional case discussed in Sect. 6.3, here the positivity condition on $a_{i,k}$ cannot be relaxed, because the transformation of sequence space coordinates described in [27] cannot be applied simultaneously to the different components of the phenotype vector.

## 7.2 Basins of attraction

It was observed in [27] that, for landscapes that satisfy the AP, the subsets and the supersets of a peak genotype combined must be contained in the *adaptive basin of attraction* (ABoA) of the peak, which is defined as the number of sequences from which the peak is accessible via fitness-monotonic paths. Since the number of elements in the subsets and the supersets of a binary sequence $\sigma$ can be written as $2^{|\sigma|} - 1$ and $2^{L-|\sigma|} - 1$ respectively, the size of the ABoA, $S_\sigma$ of $\sigma$ obeys [27]

$$S_\sigma \geq 2^{|\sigma|} + 2^{L-|\sigma|} - 2. \tag{48}$$

The existence of a lower bound on the size of ABoA's that grows exponentially with the size of the genotype space is striking and unexpected. The bound is from below since the peak could additionally be accessible from other sequences not in its sub- or supersets, and exploratory numerical simulations indicate that typical basins are often much larger than suggested by (48) [27]. Note that in the absence of any contraints, the minimal size of an ABoA is equal to the number of neighboring genotypes of a peak, which equals $L$ in the binary case.

The concept of *adaptive* basins of attraction differs importantly from the more commonly used notion of *gradient* basins, which contain those genotypes that are connected to the peak by the (unique) path of steepest ascent [16]. Gradient basins are disjoint and their union exhausts the state space, whereas adaptive basins of different peaks generally overlap. While a few previous studies have addressed the properties of gradient basins for NK fitness landscapes [41,66,67], little appears to be known about adaptive basins. A simple estimate [68] for the typical size of ABoAs for the HoC model can be derived from the scenario of accessibility percolation that was explained in Sec. 4.2. For large $L$, the fitness of a peak is close to 1 in the uniform fitness scale, and therefore the fraction of genotypes at scaled distance $\delta = d/L$ from the peak that are connected to the peak through at least one accessible path is equal to $1 - \beta_c(\delta) > 0$. Thus typical ABoAs contain a positive and principally computable fraction of all genotypes. This suggests that the recent reports of extensive adaptive basins in experimental high-throughput studies [25, 26] may in fact reflect a behavior that is generic rather than exceptional.

# 8 Conclusion

These notes have introduced probabilistic models of biological fitness landscapes, and reported recent progress in understanding their structural features. While the models are very similar, and sometimes even equivalent to the energy landscapes of interest in the statistical mechanics of disordered systems [9, 29, 61], the evolutionary perspective suggests novel research questions and brings in novel concepts. These include, for example, the quantification of accessibility through fitness-monotonic paths and adaptive basins, as well as the notion of universal epistasis, which is closely linked to sub- and supermodularity of set functions.

Let us highlight two specific scenarios of evolutionary accessibility that have been described in this text. First, in uncorrelated random landscapes, accessible paths between distant genotypes emerge through a percolation-like transition when the fitness difference between the endpoints is increased [10]. Second, structured landscapes that are submodular display a peculiar organization of accessible paths into large basins of attraction [27]. Since submodularity was argued to arise readily in fitness landscapes with an intermediate (linear) phenotype, this suggests a possible explanation for the recent observation of large-scale empirical fitness landscapes that display many peaks with large basins of attraction [25, 26] (but see the remark at the end of Section 7.2).

To conclude, we hope that these brief notes will help to motivate further work exploring cross-connections between the diverse manifestations of high-dimensional random landscapes, in statistical physics, evolutionary biology and computer science. In particular, the burgeoning field of machine learning appears to present a rich set of questions for which the concepts described here may potentially prove to be useful [69]. In this context, we note that composite maps of the form eq. (26) (typically including multiple intermediate layers) can be realized by neural network models [45, 69]. In [23] such a network was used to infer an intermediate linear phenotype and a nonlinear phenotype-fitness map from a large scale empirical data set for an essential enzyme in yeast. The composite maps in deep learning models often comprise elements that are sigmoidal rather than convex or concave, and therefore the ideas described

in Sections 6 and 7 do not obviously generalize to this setting. It is known that sigmoidal phenotype-fitness maps acting on a multi-dimensional phenotype can give rise to rugged fitness landscapes [21], but their accessibility properties remain to be explored.

## Acknowledgments

The work described here is a collaborative effort over many years. In particular, Kristina Crona, Suman Das, Jasper Franke, Muhittin Mungan, Daniel Oros and Benjamin Schmiegelt contributed essential insights and results. Special thanks are due to Peter Mörters and Jorge Peña for pointing us towards the relation to submodularity.

**Funding information** This work has been supported by Deutsche Forschungsgemeinschaft (DFG) within TRR 341 *Plant Ecological Genetics* and CRC 1310 *Predictability in Evolution*.

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
