# Peer review of "Complexity and accessibility of random landscapes"

_SciPost Physics Lecture Notes, doi:SciPost Phys. Lect. Notes 108 (2025)_

## Round 1 · Referee Report · Anonymous (Referee 1) · 2025-3-27

Strengths

1) Very clear presentation of the material, with a level of detail appropriate for some lecture notes; 2)It provides a nice introduction to this subject, that is of ongoing interest.

Report

These lecture notes provide a clear and self-consistent presentation of results on random fitness landscapes defined on discrete sequence spaces in high dimension. The study of these landscapes is primarily motivated by theoretical biology, and the notes discuss results relevant to this context, including landscape accessibility—how high-fitness genotypes can be reached through mutational paths along which fitness increases monotonically—the basins of attraction of fitness peaks, and the concept of epistasis, particularly its connection to mathematical notions such as sub-modularity. The notes also discuss links to models studied in the literature on disordered systems, such as the Random Energy Model and Hopfield models.

These notes fulfill the two criteria for publication in Scipost Lecture Notes: they provide a correct, systematic and intelligible presentation of the material, and they cover a subject of ongoing interest to the research community. The current form is already suitable for publication, in my view.

Two minor (optional) suggestions:

a) The discussion on the accessibility of landscapes satisfying universal negative epistasis in the final section is particularly interesting but inevitably slightly more technical. It could be helpful to provide an intuitive motivation for why universal negative epistasis allows one to derive results on accessibility, whereas universal positive epistasis does not. While this distinction becomes clear in the analysis of Section 7, I felt that a brief anticipatory remark in Section 6 could improve readability.

b) At the end of Section 5, the authors mention that composite genotype-fitness maps obtained by iterating Equation (26) could be interesting to explore, particularly due to their similarities with artificial neural network models. Maybe the authors could add a comment on whether the construction of sub-modular landscapes discussed in Section 6.3, as well as the discussion on the accessibility of these landscapes, could be generalized to this case.

Recommendation

Publish (meets expectations and criteria for this Journal)

---

## Round 1 · Referee Report · Anonymous (Referee 2) · 2025-9-14

Report

Report on SciPost submission arxiv:2502.05896

The lecture notes give an introduction to the subject of random fitness landscapes, motivated by evolutionary biology. The notes are very well written, authoritative, and require no specialized previous knowledge. Some of the main models used in the field are introduced and discussed. Results are derived within the text where possible, or clearly stated with references to the relevant literature. The reader is thus taken within these compact lectures, from the most basic to cutting-edge results in the field.

I therefore think these notes are a worthy contribution that meets the acceptance criteria for this journal.

I would like to suggest several places where a some additional information or context could make this review more well-rounded.

  • Given that these are lecture notes give an overview of the mathematical models and results, I feel they could benefit from giving a bit more context in the Introduction or Discussion sections. In particular, pointing the reader to relevant experimental works could be helpful--perhaps even just at the level of general reviews on the subject.

  • In the Structured Landscapes section, three models are mentioned. For one of them, Kauffman’s NK model, there is no mention of the results obtained for that model. A few sentences with references could be useful.

  • In Section 7, Eq. (42) gives a lower bound for which the authors say: “The existence of a lower bound on the size of BoA’s that grows exponentially with the size of the genotype space is striking and unexpected.” Indeed, I am aware of relatively few results for basin sizes. Can the authors comment about what else is known? For example, what is known for the HoC or rough mount Fuji models, and how does it compare with Eq. (42)?

  • On the same subject, the Discussion section states that “the evolutionary perspective suggests novel research questions” such as “the quantification of accessibility through fitness-monotonic paths.” The question of basin sizes is however also of great interest more broadly, in the fields of disordered systems and machine learning.

Recommendation

Ask for minor revision

---

## Round 2 · Author Response

We are most grateful to the reviewers for their careful reading of our manuscript, their overall positive assessment, and the suggestions for improvement and clarification. Below we provide point-by-point responses to all comments, and summarize the changes in the manuscript.

---

## Round 2 · List of Changes

Referee 1:

  1. The discussion on the accessibility of landscapes satisfying universal negative epistasis in the final section is particularly interesting but inevitably slightly more technical. It could be helpful to provide an intuitive motivation for why universal negative epistasis allows one to derive results on accessibility, whereas universal positive epistasis does not. While this distinction becomes clear in the analysis of Section 7, I felt that a brief anticipatory remark in Section 6 could improve readability.

Answer: This is a good question, and the answer is simple: Universal negative epistasis implies a result on the accessibility of local fitness peaks, whereas universal positive epistasis has analogous consequences for the accessibility of local fitness minima. We have added a corresponding statement at the end of Section 6.2.

  1. At the end of Section 5, the authors mention that composite genotype-fitness maps obtained by iterating Equation (26) could be interesting to explore, particularly due to their similarities with artificial neural network models. Maybe the authors could add a comment on whether the construction of sub-modular landscapes discussed in Section 6.3, as well as the discussion on the accessibility of these landscapes, could be generalized to this case.

Answer: We have moved the remark from the end of Section 5 to the end of the Discussion (Section 8) and elaborated on it. It is not clear to what extent the concept of submodularity can be extended to the machine learning context, as the maps used there are usually not concave or convex. On the other hand, the study of landscapes generated by sigmoidal maps is an interesting topic in its own right.

Referee 2:

  1. Given that these are lecture notes give an overview of the mathematical models and results, I feel they could benefit from giving a bit more context in the Introduction or Discussion sections. In particular, pointing the reader to relevant experimental works could be helpful–perhaps even just at the level of general reviews on the subject.

Answer: There was already some discussion of experiments on fitness landscapes in Section 3. We have now added a pointer to this work already in the introduction, and included further references.

  1. In the Structured Landscapes section, three models are mentioned. For one of them, Kauffman’s NK model, there is no mention of the results obtained for that model. A few sentences with references could be useful.

Answer: We have added brief discussions of results for the NK- and RMF-models in Section 5.

  1. In Section 7, Eq. (42) gives a lower bound for which the authors say: “The existence of a lower bound on the size of BoA’s that grows exponentially with the size of the genotype space is striking and unexpected.” Indeed, I am aware of relatively few results for basin sizes. Can the authors comment about what else is known? For example, what is known for the HoC or rough mount Fuji models, and how does it compare with Eq. (42)?

Answer: The reviewer’s question alerted us to the fact that the concept of “adaptive” basins of attraction used in this work (as well as in the experimental papers Refs.[25,26] that motivated it) differs fundamentally from the “gradient” basins that are more commonly considered especially in the physics of dynamical and disordered systems. The distinction between the two can be formalized along the lines of Ref.[16], which we have added to the references. To emphasize this important point, we now devote a separate Section 7.2. to the basins of attraction. In the newly written second paragraph of this section, we sketch an argument showing that adaptive basins in the HoC model are typically extensive, in the sense that they contain a nonzero fraction of all genotypes. This argument will be worked out fully in an upcoming publication.

  1. On the same subject, the Discussion section states that “the evolutionary perspective suggests novel research questions” such as “the quantification of accessibility through fitness-monotonic paths.” The question of basin sizes is however also of great interest more broadly, in the fields of disordered systems and machine learning.

Answer: We fully agree and now mention the adaptive basins (discussed in detail in Section 7.2, see preceding comment) explicitly in Section 8

Additional changes: References [2,3,16,21,53,62,64,65,66,67] have been added, and a few minor typos have been corrected.

---

## Editorial Decision

published